# MISIM: A NOVEL CODE SIMILARITY SYSTEM

## ABSTRACT

Semantic code similarity systems are integral to a range of applications from code recommendation to automated software defect correction. Yet, these systems still lack the maturity in accuracy for general and reliable wide-scale usage. To help address this, we present *Machine Inferred Code Similarity* (MISIM), a novel end-to-end code similarity system that consists of two core components. First, MISIM uses a novel *context-aware semantic structure* (CASS), which is designed to aid in lifting semantic meaning from code syntax. We compare CASS with the abstract syntax tree (AST) and show CASS is more accurate than AST by up to $1.67\times$. Second, MISIM provides a neural-based code similarity scoring algorithm, which can be implemented with various neural network architectures with learned parameters. We compare MISIM to four state-of-the-art systems: *(i)* Aroma, *(ii)* code2seq, *(iii)* code2vec, and *(iv)* Neural Code Comprehension, . In our experimental evaluation across 328,155 programs (over 18 million lines of code), MISIM has $1.5\times$ to $43.4\times$ better accuracy across all four systems.

## 1 INTRODUCTION

The field of *machine programming* (MP) is concerned with the automation of software development (Gottschlich et al., 2018). In recent years, there has been an emergence of many MP systems, due, in part, to advances in machine learning, formal methods, data availability, and computing efficiency (Allamanis et al., 2018a; Alon et al., 2018; 2019b;a; Ben-Nun et al., 2018; Cosentino et al., 2017; Li et al., 2017; Luan et al., 2019; Odena & Sutton, 2020; Tufano et al., 2018; Wei & Li, 2017; Zhang et al., 2019; Zhao & Huang, 2018). An open challenge in MP is the construction of accurate *code similarity* systems. *Code similarity* is the problem of determining if two or more code snippets have some degree of semantic similarity (or equivalence) even in the presence of syntactic divergence. At the highest level, code similarity systems aim to determine if two or more code snippets are solving a similar problem, even if the implementations they use differ (e.g., various algorithms of `sort()` (Cormen et al., 2009)).

Historically, code similarity systems have been considered an auxiliary feature that aim to improve programmer productivity with tools such as code recommendation, automated bug detection, and language-to-language transformation for small kernels (e.g., stencils), to name a few (Allamanis et al., 2018b; Ahmad et al., 2019; Bader et al., 2019; Barman et al., 2016; Bhatia et al., 2018; Dinella et al., 2020; Kamil et al., 2016; Luan et al., 2019; Pradel & Sen, 2018). Yet, these systems still lack the maturity in accuracy for general and reliable wide-scale usage. In particular, without largely accurate code similarity systems to automate significant parts of our software development, we believe the explosion of heterogeneous software and hardware may become an untenable problem that software developers will not be able to navigate (Ahmad et al., 2019; Batra et al., 2018; Bogdan et al., 2019; Chen et al., 2020; Deng et al., 2020; Hannigan et al., 2019) . Yet, as others have noted before us, even some of the most fundamental questions in code similarity have no clear answers, such as the proper structural representation of code for a particular similarity problem (Alam et al., 2019; Allamanis et al., 2018b; Becker & Gottschlich, 2017; Ben-Nun et al., 2018; Dinella et al., 2020; Iyer et al., 2020; Luan et al., 2019). In this paper, we aim to address some of these questions.

While prior work has explored some structural representations of code in the space of code similarity, these explorations are far from complete. The abstract syntax tree (AST) is used in the code2vec and code2seq system (Alon et al., 2019b;a), a novel structure called the contextual flow graph (XFG) is used by Neural Code Comprehension (NCC) (Ben-Nun et al., 2018), and a new structure called the simplified parse tree (SPT) is used by Aroma (Luan et al., 2019). While each of these representations have benefits in certain contexts, we have found that they possess one or more limitations when used

in the broader context of code similarity that may limit their practical application. For example, the AST – while principally valuable for compilers – is syntax driven, which can often mislead code similarity systems into learning too much syntax and not enough semantics (i.e., the meaning behind the syntax). The XFG is obtained from an intermediate representation (IR), which requires code compilation; this limits its use to only compilable code. Although the SPT is structurally driven (not syntax driven), it does not always resolve syntactic ambiguities, which may result in semantic obfuscation that prevent it from observing semantic variation caused by contextual syntactic differences. In Section 4, we evaluate how these limitations impact code similarity accuracy.

Learning from these observations, we attempt to address some of the open questions around code similarity with our novel end-to-end code similarity system called _Machine Inferred Code Similarity_ (MISIM). In this paper, we principally focus on two main novelties of MISIM and how they may improve code similarity analysis: _(i)_ its structural representation of code, called the _context-aware semantic structure_ (CASS), and _(ii)_ its neural-based _learned_ code similarity scoring algorithm. These components can be used individually or together as we have chosen to do.

This paper makes the following technical contributions:

- We present _Machine Inferred Code Similarity_ (MISIM), a novel end-to-end code similarity system.
- We present MISIM's _context-aware semantic structure_ (CASS), a novel structural representation of code specifically designed to _(i)_ lift semantic meaning from code syntax and _(ii)_ provide an extensible representation that can be augmented as needed (e.g., as new programming languages (PLs) emerge, existing PL contextual syntax ambiguities are discovered, etc.). We also experiment AST and CASS on representing code semantics. Our preliminary result shows that CASS can be up to $1.67\times$ more accurate.
- We present MISIM's open-ended deep neural network (DNN) backend that _learns_ the similarity scoring algorithm for a given code corpus and show its efficacy across three DNN topologies: _(i)_ bag-of-features, _(ii)_ a recurrent neural network (RNN), and _(iii)_ a graph neural network (GNN). [1]
- We compare MISIM to four state-of-the-art code similarity systems: _(i)_ code2vec, _(ii)_ code2seq, _(iii)_ Neural Code Comprehension, and _(iv)_ Aroma. Our experimental evaluation, across 328,155 C/C++ programs comprising of over 18 million lines of code, illustrates that MISIM is more accurate than all four systems, _across all experiments_, ranging from $1.5\times$ to $43.4\times$.

## 2 MISIM SYSTEM

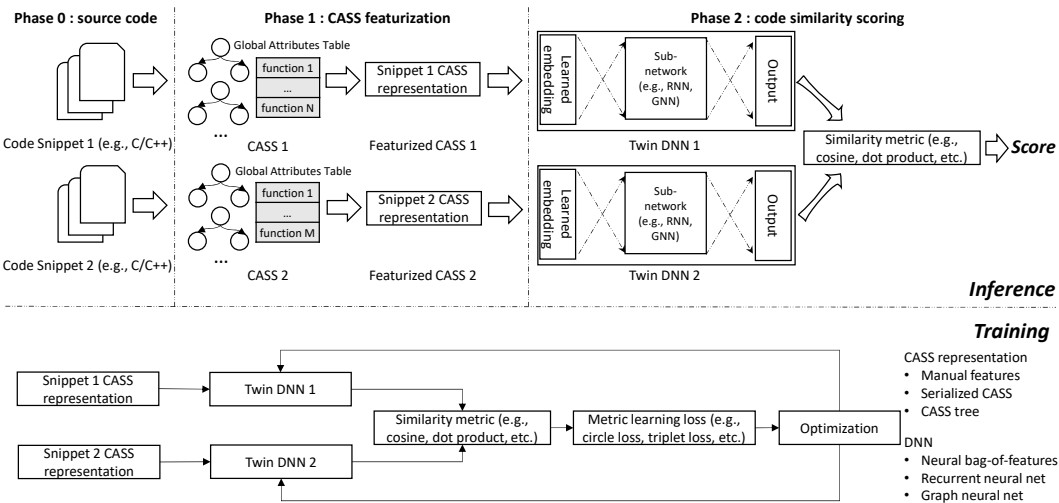

Figure 1: Overview of the MISIM System.

---

[1]We acknowledge that this is a non-exhaustive list of all possible neural network architectures that can be used with MISIM. We explore these three for two reasons: _(i)_ to demonstrate the diversity of DNNs that can be used with MISIM and _(ii)_ to illustrate how different DNNs have a measurable impact on MISIM's accuracy.

In Figure 1, we provide an overview of MISIM's system diagram. A core component of MISIM is the novel *context-aware semantic structure* (CASS), which aims to capture semantically salient properties of the input code. Moreover, CASS is designed to be *context-aware*. That is, it can capture information that describes the context of the code (e.g., parenthetical operator disambiguation between a function call, mathematical operator precedence, Boolean logic ordering, etc.) that may otherwise be ambiguous without such context-sensitivity. Once these CASSes are constructed, they are vectorized and used as input to a neural network, which produces a feature vector for a corresponding CASS. Once a feature vector is generated, a code similarity measurement (e.g., cosine similarity (Baeza-Yates & Ribeiro-Neto, 1999)) calculates the similarity score.

## 2.1 CONTEXT-AWARE SEMANTIC STRUCTURE (CASS)

We have designed CASS with the following guiding principles: *(i)* it should not require compilation, *(ii)* it should be a flexible representation that captures code semantics, and *(iii)* it should be capable of resolving code ambiguities in both its context sensitivity to the code and its environment. The first principle *(i)* originates from the observation that unlike programs in higher-level scripting programming languages (e.g., Python (Van Rossum & Drake, 2009), JavaScript (Flanagan, 2006)), C/C++ programs found "in the wild" may not be well-formed (e.g., due to specialized compiler dependencies) or exhaustively include all of their dependencies (e.g., due to assumptions about library availability) and therefore may not compile. Moreover, for code recommendation systems that are expected to function in a live setting, requiring compilation may severely constrain their practical application. We address this by introducing a structure such that it does not require compilation (Section 2.1.1). The second *(ii)* and third *(iii)* principles originate from the observation that different scenarios may require attention to different semantics (e.g., embedded memory-bound systems may prefer to use algorithms that do not use recursion due to a potential call stack overflow) and that programming languages (PLs) evolve and new PLs continue to emerge. We attempt to address these issues with CASS's configuration categories (Section 2.1.2).

### 2.1.1 CASS'S TREES AND GLOBAL ATTRIBUTES TABLE

In this section, we provide an informal definition of CASS (a formal definition can be found in Appendix A). The CASS consists of one or more CASS trees and an optional global attributes table (GAT). A CASS tree is a tree, in which the root node represents the entire span of the code snippet. During the construction of a CASS tree, the program tokens are mapped to its corresponding node label using the grammar of the high-level programming language. A CASS's GAT contains exactly one entry per unique function definition in the code snippet. A GAT entry currently includes only the input and output cardinality values for each corresponding function, but can be extended as new global attributes are needed. A simple example of this is illustrated in Figure 2, where a code snippet and its corresponding CASS tree and GAT are shown.

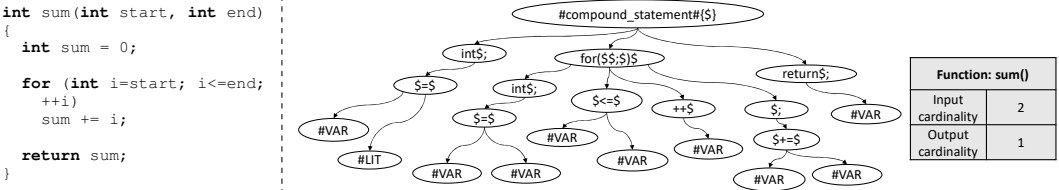

Figure 2: A Sum Function and One Variant of Its Context-Aware Semantic Structure.

### 2.1.2 CASS CONFIGURATION CATEGORIES

In general, CASS configurations can be broadly classified into two categories: *language-specific* and *language-agnostic*. Options for each of the configuration categories are described in Table 1.

**Language-specific configurations (LSCs).** Language-specific configurations are meant to capture semantic meaning by resolving a syntactic ambiguity present in the concrete syntax tree. It also introduces specificity related to the high-level programming language. For example, the parentheses operator is overloaded in many programming languages to enforce an order of evaluation of operands in an expression as well as to enclose a list of function arguments, amongst other things. CASS

Table 1: CASS Configuration Options.

| Category | Type | Option |
|---|---|---|
| Language-specific | A. Node Prefix Label | 0. No change (Aroma's original configuration)
1. Add a prefix to each internal node label
2. Add a prefix to parenthesis node label (C/C++ Specific) |
| Language-agnostic | B. Compound Statements | 0. No change (Aroma's original configuration)
1. Remove all features relevant to compound statements
2. Replace with '{#}' |
| | C. Global Variables | 0. No change (Aroma's original configuration)
1. Remove all features relevant to global variables
2. Replace with '#GVAR'
3. Replace with '#VAR' (the label for local variables) |
| | D. Global Functions | 0. No change (Aroma's original configuration)
1. Remove all features relevant to global functions
2. Remove function identifier and replace with '#EXFUNC' |
| | E. Function I/O Cardinality | 0. No change
1. Include the input and output cardinality per function |

disambiguates these by explicitly embedding the semantic contextual information in the CASS tree nodes using the *node prefix label* (defined in Appendix A).

*(A) Node Prefix Label.* The configuration options for node prefix labels[2] correspond to various levels of semantic to syntactic information. In Table 1, option 0 corresponds to the extreme case of a concrete syntax embedding, option 1 corresponds to eliminating irrelevant syntax, and option 2 is principally equivalent to option 1, except it applies only to parentheticals, which we have identified – through empirical evaluation – to often have notably divergent semantic meaning based on context.

**Language-agnostic configurations (LACs).** LACs can improve code similarity analysis by unbinding overly-specific semantics that may be present in the original concrete syntax tree structure.

*(B) Compound Statements.* The *compound statements* configuration option enables the user to control how much non-terminal node information is incorporated into the CASS. Option 0 is equivalent to Aroma's SPT, option 1 omits separate features for compound statements altogether, and option 2 does not discriminate between compound statements of different lengths and specifies a special label to denote the presence of a compound statement.

*(C) Global Variables.* The *global variables* configuration specifies the degree of global variable-specific information contained in a CASS. The configuration options for global variables provide the user with the ability to control the level of abstraction – essentially binding or unbinding global variable names as needed. If all code similarity analysis will be performed against the same software program, retaining global variable names may help elicit deeper semantic meaning. If not, unbinding global variable names may improve semantic meaning.

*(D) Global Functions.* The *global functions* configuration serves the dual purpose of *(i)* controlling the amount of function-specific information to featurize and *(ii)* to explicitly disambiguate between the usage of global functions and global variables (a feature that is absent in Aroma's SPT design).

*(E) Function I/O Cardinality.* The *function I/O cardinality* configuration aims to abstract the semantics of certain groups of functions through input and output cardinality (i.e., embedded semantic information that can be implicitly derived by analyzing the number of input and output parameters of a function).

We have found that the specific context in which code similarity is performed seems to provide some indication of the optimal specificity of the CASS configuration. In other words, one specific CASS configuration is unlikely to work in all scenarios. To address this, CASS provides a number of options to control the language-specific and/or language-agnostic configurations to enable tailored CASS representations for particular application contexts. We discuss this in greater detail in Appendix A.1.

### 2.1.3   CASS VS AST

Some recent research on code representation uses the AST-based representation (Dinella et al., 2020) or AST paths (Alon et al., 2019b;a). In this subsection, we explore how AST and CASS perform on the task of code semantic representation described here.

---

[2]Analytically deriving the optimal selection of node prefix labels across all C/C++ code may be untenable. To accommodate this, we currently provide two levels of granularity for C/C++ node prefix labels in CASS.

We compared the code similarity performance of ASTs and CASSes on the test set of POJ-104, as shown in table 3, by transforming both kinds of representations into feature vectors. using the same method described in (Luan et al., 2019) and compute the similarity scores using dot or cosine similarity. For each program in the dataset, we extracted its CASS under three different configurations: 0-0-0-0-0 [3], the base configuration, and 2-1-3-1-1/1-2-1-0-0, the best/worst performing configuration according to our preliminary evaluation of CASS (see Appendix C.5 for details). We also extracted the ASTs of function bodies in a program. Each syntax node in the AST is labeled by its node type, and an identifier (or literal) node also gets a single child labeled by the corresponding identifier name (or literal text).

Table 2: Test Accuracy for AST and three CASS configurations on POJ-104.

| Method | MAP@R (%) | AP (%) | AUPRG (%) |
|---|---|---|---|
| AST-Dot | 45.12 | 35.98 | 97.29 |
| AST-Cos | 47.39 | 45.31 | 98.41 |
| CASS (0-0-0-0-0)-Dot | 52.09 | 45.99 | 98.42 |
| CASS (0-0-0-0-0)-Cos | 55.12 | 55.4 | 99.07 |
| CASS (2-1-3-1-1)-Dot | 55.59 | 48.31 | 98.62 |
| CASS (2-1-3-1-1)-Cos | **60.78** | **60.42** | **99.31** |
| CASS (1-2-1-0-0)-Dot | 52.74 | 40.73 | 97.87 |
| CASS (1-2-1-0-0)-Cos | 57.99 | 54.75 | 99.06 |

As shown in Table 2, CASS configurations show an improvement in accuracy over the AST for up to $1.67\times$ across three evaluation metrics described in Appendix C.3. To better understand the performance difference, we investigated a few solutions for the same problems from the POJ-104 dataset. One of the interesting observations we found is that for the same problem, a solution may have a different naming convention for local variables than that of another solution (e.g., English vs Mandarin description of variables), but the resulting different variable names may carry the same semantic meaning. AST uses variable names in its structure, but CASS has the option to not use variable names. Thus the erasure of local variable names in CASS might help in discovering the semantic similarity between code with different variable names. This might explain some of the performance difference between AST and CASS in this experiment.

## 2.2 NEURAL SCORING ALGORITHM

MISIM's neural scoring algorithm aims to compute the similarity score of two input programs. The algorithm consists of two phases. The first phase involves a neural network model that maps a featurized CASS to a real-valued code vector. The second phase generates a similarity score between a pair of code vectors using a similarity metric.[4] For the remainder of this section, we describe the details of the scoring model, its training strategy, and other neural network model choices.

### 2.2.1 MODEL

We investigated three neural network approaches for MISIM's scoring algorithm: *(i)* a graph neural network (GNN) (Zhou et al., 2018), *(ii)* a recurrent neural network (RNN), and *(iii)* a bag of manual features (BoF) neural network. We name these models MISIM-GNN, MISIM-RNN, and MISIM-BoF respectively. MISIM-GNN performs the best overall for our experiments, therefore, we describe it in detail in this section. Appendix B has details of the MISIM-RNN and MISIM-BoF models.

**MISIM-GNN.** MISIM-GNN's architecture is shown in Figure 3. For this approach, an input program's CASS representation is transformed into a graph. Then, each node in the graph is embedded into a trainable vector, serving as the node's initial state. Next, a GNN is used to update each node's state iteratively. Finally, a global readout function is applied to extract a vector representation of the entire graph from the nodes' final states. We describe each of these steps in more detail below.

**Input Graph Construction.** We represent each program as a single CASS instance. Each instance can contain one or more CASS trees, where each tree corresponds to a unique function of the program.

---

[3]Configuration 0-0-0-0-0 is the duplicate of SPT. As shown in Table 2, configuration 2-1-3-1-1 shows better accuracy than configuration 0-0-0-0-0.

[4]For this work, we have chosen cosine similarity as the similarity metric used within MISIM.

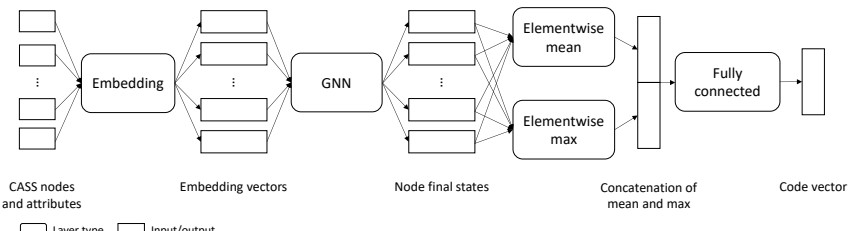

Figure 3: MISIM-GNN Architecture.

The CASS instance is converted into a single graph representation to serve as the input to the model. The graph is constructed by first transforming each CASS tree and its GAT entry into an individual graph. These graphs are then merged into a single (disjoint) graph. For a CASS consisting of a CASS tree $T = (V, E)$ and a GAT entry $a$, we transform it into a directed graph $G = (V', E', R)$, where $V'$ is the set of graph nodes, $R$ is the set of edge types, and $E' = \{(v, u, r) \mid v, u \in V', r \in R\}$ is the set of graph edges. The graph is constructed as follows:

$$V' = V \cup \{a\}, \qquad R = \{p, c\}, \qquad E' = \{(v, u, p) \mid (v, u) \in E\} \cup \{(v, u, c) \mid (u, v) \in E\}.$$

The two edge types, $p$ and $c$, represent edges from CASS tree nodes to their child nodes and to their parent nodes, respectively.

**Graph Neural Network.** MISIM embeds each node $v \in V'$ in the input graph $G$ into a vector by assigning a trainable vector to each unique node label (with the optional prefix) and GAT attribute. The node embeddings are then used as node initial states ($\mathbf{h}_v^{(0)}$) by a relational graph convolutional network (R-GCN (Schlichtkrull et al., 2018)) specified as the following:

$$\mathbf{h}_v^{(l)} = \text{ReLU}\left( \frac{1}{\sum_{r \in R} |\mathcal{N}_v^r|} \sum_{r \in R} \sum_{u \in \mathcal{N}_v^r} \mathbf{W}_r^{(l)} \mathbf{h}_u^{(l-1)} + \mathbf{W}_0^{(l)} \mathbf{h}_v^{(l-1)} \right) \qquad v \in V', l \in [1, L],$$

where $L$ is the number of GNN layers, $\mathcal{N}_v^r = \{u \mid (u, v, r) \in E'\}$ is the set of neighbors of $v$ that connect to $v$ through an edge of type $r \in R$, and $\mathbf{W}_r^{(l)}, \mathbf{W}_0^{(l)}$ are weight matrices to be learned.

**Code Vector Generation.** To obtain a vector representing the entire input graph, we apply both an average pooling and a max pooling on the graph nodes' final states ($\mathbf{h}_v^{(L)}$). The resulting two vectors are concatenated and fed into a fully-connected layer, yielding the code vector for the input program.

### 2.2.2 TRAINING

We train the neural network model following the setting of metric learning (Schroff et al., 2015; Hermans et al., 2017; Musgrave et al., 2020; Sun et al., 2020), which tries to map input data to a vector space where, under a distance (or similarity) metric, similar data points are close together (or have large similarity scores) and dissimilar data points are far apart (or have small similarity scores). The metric we use is the cosine similarity in the code vector space. As shown in the lower half of Figure 1, we use pair-wise labels to train the model. Each pair of input programs are mapped to two code vectors by the model, from which a similarity score is computed and optimized using a metric learning loss function.

## 3 RELATED WORK

Although research in code similarity is still in early stages, there is a growing body of exploratory work (Alon et al., 2019b;a; Ben-Nun et al., 2018; Kim et al., 2018; Liu et al., 2018; Luan et al., 2019). In this section, we briefly discuss four state-of-the-art code similarity systems: code2vec, code2seq, Neural Code Comprehension (NCC), and the Aroma system.

*code2vec and code2seq.* A core goal of code2vec is to learn a *code embedding* for representing snippets of code. A code embedding is trained through the task of *semantic labeling of code snippets* (i.e., predicting the function name for a function body). code2seq is proposed as a approach to generate a natural language sequence from a code snippet by encoding the code into vectors and

decoding a sequence from them, where the sequence can be a description of the semantics of the code (e.g., the function name). As input, both systems use abstract syntax tree (AST) paths to represent a code snippet. The AST is a tree structure that represents the syntactic information of the source code (Baxter et al., 1998). code2vec uses a fixed vocabulary of paths to map each path into a trainable vector, while code2seq encodes the paths with an RNN. They incorporate attention-based neural networks to automatically identify AST paths that are more relevant to deriving code semantics. In contrast, the MISIM system uses the CASS structure that provides abstractions over code syntax, and a neural backend that can be instantiated by various learning algorithms.

*Neural Code Comprehension (NCC).* NCC attempts to learn code semantics based on an intermediate representation (IR) (Lattner & Adve, 2004) of the code. NCC processes source code at the IR level in an attempt to extract additional semantic meaning. It transforms IR into a *contextual flow graph* (XFG) that incorporates both data- and control-flow of the code, and uses XFG to train a neural network for learning a *code embedding*. One of the constraints of NCC is that its IR requires code compilation. MISIM does not use an IR and thus does not have a compilation requirement, which may be helpful in some settings (e.g., live programming environments).

*Aroma.* Aroma is a code recommendation system that takes a partially-written code snippet and recommends extensions for the snippet. The intuition behind Aroma is that programmers often write code that may have already been written. Aroma leverages a code base of functions and recommends extensions in a live setting. Aroma introduces the simplified parse tree (SPT) - a tree structure to represent a code snippet. Unlike an AST, an SPT is not language-specific and enables code similarity comparison across various programming languages. To compute the similarity score of two code snippets, Aroma extracts binary feature vectors from SPTs and calculate their dot product.

*Other Types of Code Comprehension.* There is also a growing body of work on code comprehension that is not directly intended for code similarity, but may provide indirect value to the space of code similarity. A large body of work has studied applying machine learning to learn from the AST for completing various tasks on code (Alon et al., 2019a; Chen et al., 2018; Hu et al., 2018; Li et al., 2018; Mou et al., 2016). Odena & Sutton (2020) represent a program as property signatures inferred from input-output pairs, which can be used to improve program synthesizers, amongst other things. There has also been work exploring graph representations, such as the following. For bug detection and code generation, Allamanis et al. (2018b); Brockschmidt et al. (2019) represent a program as a graph with a backbone AST and additional edges representing lexical ordering and semantic relations between the nodes. Dinella et al. (2020) also use a AST-backboned graph representation of programs to learn bug fixing through graph transformation. Hellendoorn et al. (2020) introduce a simplified graph containing only AST leaf nodes for program repair. Wei et al. (2020) extract type dependency graphs from JavaScript programs for probabilistic type inference. In this paper, we present a new code structural representation that can be consumed by different neural architectures to infer the semantic similarity of code.

## 4 EXPERIMENTAL EVALUATION

In this section, we analyze the performance of MISIM compared to code2vec, code2seq, NCC, and Aroma on a dataset containing more than 328,000 programs. Overall, we find that MISIM has improved performance than these systems across three metrics. Then, we compare the performance of two MISIM variants, each trained with a different CASS configuration.

Table 3: Dataset statistics.

| Split | GCJ | | POJ-104 | |
| --- | --- | --- | --- | --- |
| | #Problems | #Programs | #Problems | #Programs |
| Training | 237 | 223,171 | 64 | 28,137 |
| Validation | 29 | 36,409 | 16 | 7,193 |
| Test | 31 | 22,795 | 24 | 10,450 |
| Total | 297 | 282,375 | 104 | 45,780 |

**Datasets.** Our experiments are conducted on two datasets: the GCJ dataset (Ullah et al., 2019) and the POJ-104 dataset (Mou et al., 2016). The GCJ dataset consists of solutions to programming problems in Google's Code Jam coding competitions[5]. We use a subset of it that only contains solutions written in C/C++, solving 297 problems. The POJ-104 dataset consists of student-written C/C++ programs solving 104 problems. For both datasets, we label two programs as similar if they are solutions to the same problem. After a filtering step, which removes unparsable/non-compilable programs, we split

---

[5]https://codingcompetitions.withgoogle.com/codejam

each dataset by problem into three subsets for *training*, *validation*, and *testing*. Detailed statistics of the dataset partitioning are shown in Table 3.

**Training.** Unless otherwise specified, we use the same training procedure in all experiments. The models are built and trained using PyTorch (Paszke et al., 2019). To train the models, we use the Circle loss (Sun et al., 2020), a state-of-the-art metric learning loss function that has been tested effective in various similarity learning tasks. Following the P-K sampling strategy (Hermans et al., 2017), we construct a batch of programs by first randomly sampling 16 different problems, and then randomly sampling at most 5 different solutions for each problem. The loss function takes the similarity scores of all intra-batch pairs and their pair-wise labels as input. Further details about the training procedure and hyperparameters are discussed in Appendix C.1.

**Evaluation Metrics.** The accuracy metrics we use for evaluation are Mean Average Precision at R (MAP@R) (Musgrave et al., 2020), Average Precision (AP) (Baeza-Yates & Ribeiro-Neto, 1999), and Area Under Precision-Recall-Gain Curve (AUPRG) (Flach & Kull, 2015). Since these metrics are already defined, we do not detail them here, but would refer readers to Appendix C.3.

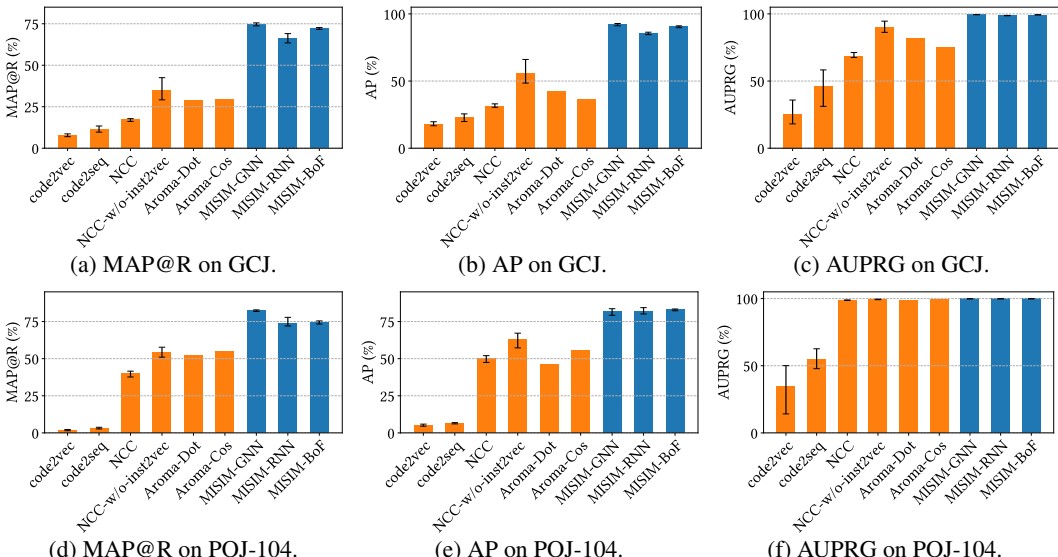

(a) MAP@R on GCJ.   (b) AP on GCJ.   (c) AUPRG on GCJ.

(d) MAP@R on POJ-104.   (e) AP on POJ-104.   (f) AUPRG on POJ-104.

Figure 4: Summarized accuracy results on the test sets for code2vec, code2seq, NCC, and Aroma and MISIM. Bar heights are the averages of the measurements over 3 runs, and error bars are bounded by the minimum and the maximum of measured values.

**Configuration Identifier.** In the following sections, we refer to a configuration of CASS by its unique identifier (ID). A configuration ID is formatted as A-B-C-D-E. Each of the five letters corresponds to a configuration type in the second column of Table 1, and will be replaced by an option number specified in the third column of the table. Configuration 0-0-0-0-0 corresponds to Aroma's SPT.

**Results.** Figure 4 shows the accuracy of MISIM, code2vec, code2seq, NCC, and Aroma[6]. The blue bars show the results of the MISIM system variants trained using the baseline CASS configuration of 0-0-0-0-0. The orange bars show the results of code2vec, code2seq, NCC, and Aroma. We observe that MISIM-GNN results in the best performance for MAP@R, yielding 1.5× to 43.4× improvements over the other systems. In some cases, MISIM-BoF achieves the best AP and AUPRG scores. In summary, MISIM system has better accuracy than the other systems we compared against across all three metrics.

**Results Analysis.** We provide a brief intuition for these results. The code2vec and code2seq systems use paths in an abstract syntax tree (AST), a purely syntactical code representation, as the input to its neural network. We speculate that such representation may *(i)* keep excessive fine-grained syntactical details while *(ii)* omitting structural (i.e., semantic) information. This may explain why code2vec and code2seq has smaller accuracy in our experiments. The Aroma system employs manual features derived from the simplified parse tree and computes the number of overlapping features from two

---

[6]The table illustrating the same results, but shown numerically, can be found in Appendix C.4.

programs as their similarity score. The selection of manual features appears to be heuristic-based and might potentially result in a loss in semantic information. NCC tries to learn code semantics from LLVM IR, a low level code representation designed for compilers. The lowering process from source code to LLVM IR may discard some semantic-relevant information such as identifier names and syntactic patterns, which is usually not utilized by compilers, but might be useful for inferring code semantics. The absence of such information from NCC's input may limit its code similarity accuracy.

Table 4: Test Accuracy of MISIM-GNN Trained on Different Subsets of the Training Set. The results are shown as the average and min/max values relative to the average over 3 runs.

| Sub-Training Set | Configuration | MAP@R (%) | AP (%) | AUPRG (%) |
|---|---|---|---|---|
| $T_A$ | $C_1$ | 69.78 (-0.42/+0.21) | 76.39 (-1.68/+1.51) | 99.78 (-0.03/+0.03) |
| | $C_2$ | **71.99 (-0.26/+0.45)** | **79.89 (-1.20/+0.71)** | **99.83 (-0.02/+0.01)** |
| $T_B$ | $C_1$ | 63.45 (-1.58/+1.92) | 68.58 (-2.51/+2.85) | 99.63 (-0.06/+0.06) |
| | $C_2$ | **67.40 (-1.85/+1.23)** | **69.86 (-3.34/+1.79)** | **99.65 (-0.10/+0.05)** |
| $T_C$ | $C_1$ | **63.53 (-1.08/+1.53)** | **72.47 (-0.95/+1.24)** | **99.70 (-0.04/+0.03)** |
| | $C_2$ | 61.23 (-2.04/+1.57) | 69.83 (-1.03/+1.60) | 99.65 (-0.03/+0.03) |
| $T_D$ | $C_1$ | **61.78 (-0.46/+0.47)** | **66.86 (-2.31/+2.81)** | **99.56 (-0.06/+0.07)** |
| | $C_2$ | 60.86 (-1.59/+0.90) | 63.86 (-3.06/+3.43) | 99.46 (-0.14/+0.11) |

## 4.1 SPECIALIZED EXPERIMENTAL RESULT: CASS CONFIGURATIONS

In this subsection, we provide early anecdotal evidence indicating that no CASS configuration is invariably the best for all code snippets. In other words, the configurations may need to be chosen based on the characteristics of code that MISIM will be trained on and, eventually, used for. We conducted a series of experiments that train MISIM-GNN models with two CASS configurations on several randomly sampled sub-training sets and compared their test accuracy. The two configurations used were $C_1$, which is the CASS baseline configuration of 0-0-0-0-0 and $C_2$, which is a non-baseline configuration of 2-2-3-2-1 that provides a higher abstraction over the source code than $C_1$ (e.g., replace global variable names with a unified string, etc.). Table 4 shows the results from four selected sub-training sets, named $T_A$, $T_B$, $T_C$, and $T_D$ from POJ-104. It can be seen that when trained on $T_A$ or $T_B$, the system using configuration $C_2$ performs better than the baseline configuration in all three accuracy metrics. However, using the training sets $T_C$ or $T_D$, the results are inverted.

To better understand this divergence, we compared the semantic features of $T_A \cap T_B$ to $T_C \cap T_D$. We observed that some CASS-defined semantically salient features (e.g., global variables – see Appendix A) that $C_2$ had been customized to extract, occurred less frequently in $T_A \cap T_B$ than in $T_C \cap T_D$. We speculate that, in the context of the POJ-104 dataset, when global variables are used more frequently, they are more likely to have consistent meaning across different programs. As a result, abstracting them away as $C_2$ does for $T_C, T_D$, leads to a loss in semantic information salient to code similarity. Conversely, when global variables are not frequently used, there is an increased likelihood that the semantics they extract are specific to a single program's embodiment. As such, retaining their names in a CASS, may increase syntactic noise, thereby reducing model performance. Therefore, when $C_2$ eliminates them for $T_A, T_B$, there is an improvement in accuracy.

## 5 CONCLUSION

In this paper, we presented MISIM, an end-to-end code similarity system. MISIM has two core novelties. The first is the *context-aware semantics structure* (CASS) designed specifically to lift semantic meaning from code syntax. The second is a neural-based code similarity scoring algorithm for learning code similarity scoring using CASS. Our experimental evaluation showed that MISIM outperforms four other state-of-the-art code similarity systems usually by a large factor (up to 43.4×). We also provided anecdotal evidence illustrating that there may not be one universally optimal CASS configuration. An open research question for MISIM is in how to automatically derive the proper configuration of its various components for a given code corpus, specifically the CASS and neural scoring algorithms which we plan to explore in future work.

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

# A  CONTEXT-AWARE SEMANTICS STRUCTURE DETAILS

The following is the formal definition of CASS.

**Definition 1 (Context-aware semantic structure (CASS))** *A CASS consists of one or more CASS trees and an optional global attributes table (GAT). A CASS tree, $T$, is a collection of nodes, $V = \{v_1, v_2, \ldots, v_{|V|}\}$, and edges, $E = \{e_1, e_2, \ldots, e_{|E|}\}$, denoted as $T = (V, E)$. Each edge is directed from a parent node, $v_p$ to a child node, $v_c$, or $e_k = (v_p, v_c)$ where $e_k \in E$ and $v_p, v_c \in V$. The root node, $v_r$, of the tree signifies the beginning of the code snippet and has no parent node, i.e., $\nexists v_p, (v_p, v_r) \in E$. A child node is either an internal node or a leaf node. An internal node has at least one child node while a leaf node has no child nodes. A CASS tree can be empty, in which it has no nodes. The CASS GAT contains exactly one entry per unique function definition in the code snippet. A GAT entry includes the input and output cardinality values for the corresponding function.*

**Definition 2 (Node labels)** *Every CASS node has an associated label, $l_v$. During the construction of a CASS tree, the program tokens at each node, $t_v$ are mapped to its corresponding label or $l_v = f(t_v)$. This is depicted with an expression grammar for node labels and the function mapping tokens to labels below.[7]*

$\langle bin\text{-}op \rangle$        ::= '+' | '−' | '⋆' | '/' | ...

$\langle unary\text{-}op \rangle$      ::= '++' | '−−' | ...

$\langle leaf\text{-}node\text{-}label \rangle$    ::= LITERAL | IDENT | '#VAR' | '#GVAR' | '#EXFUNC' | '#LIT' | ...

$\langle exp \rangle$            ::= '$' | '$' $\langle bin\text{-}op \rangle$ '$' | $\langle unary\text{-}op \rangle$ '$' | ...

$\langle internal\text{-}node\text{-}label \rangle$ ::= 'for' '(' $\langle exp \rangle$ ';' $\langle exp \rangle$ ';' $\langle exp \rangle$ ')' $\langle exp \rangle$ ';'
                | 'int' $\langle exp \rangle$ ';'
                | 'return' $\langle exp \rangle$ ';'
                | $\langle exp \rangle$
                | ...

$$l_v = f(t_v) = \begin{cases} \langle leaf\text{-}node\text{-}label \rangle & \text{if } v \text{ is a leaf node} \\ \langle internal\text{-}node\text{-}label \rangle & \text{otherwise} \end{cases}$$

**Definition 3 (Node prefix label)** *A node prefix label is a string prefixed to a node label. A node prefix label may or may not be present.*

## A.1  DISCUSSION

We believe there is no silver bullet solution for code similarity for all programs and programming languages. Based on this belief, a key intuition of CASS's design is to provide a structure that is semantically rich based on structure, with inspiration from Aroma's SPT, while simultaneously providing a range of customizable parameters to accommodate a wide variety of scenarios. CASS's language-agnostic and language-specific configurations and their associated options serve for exploration of a series of tree variants, each differing in their granularity of detail of abstractions.

For instance, the *compound statements* configuration provides three levels of abstraction. Option 0 is Aroma's baseline configuration and is the finest level of abstraction, as it featurizes the number of constituents in a compound statement node. Option 2 reduces compound statements to a single token and represents a slightly higher level of abstraction. Option 1 eliminates all features related to compound statements and is the coarsest level of abstraction. The same trend applies to the *global variables* and *global functions* configurations. It is our belief, based on early evidence, that the appropriate level of abstraction in CASS is likely based on many factors such as *(i)* code similarity purpose, *(ii)* programming language expressiveness, and *(iii)* application domain.

Aroma's original SPT seems to work well for a common code base where global variables have consistent semantics and global functions are standard API calls also with consistent semantics (e.g., a single code-base). However, for cases outside of such spaces, some question about applicability

---

[7]Note: the expression grammar we provide is non-exhaustive due to space limitations. The complete set of standard C/C++ tokens or binary and unary operators is collectively denoted in shorthand as '...'.

arise. For example, assumptions about consistent semantics for global variables and functions may not hold in cases of non-common code-bases or non-standardized global function names (Wulf & Shaw, 1973; Gellenbeck & Cook, 1991; Feitelson et al., 2020). Having the capability to differentiate between these cases, and others, is a key motivation for CASS.

We do not believe that CASS's current structure is exhaustive. With this in mind, we have designed CASS to be extensible, enabling a seamless mechanism to add new configurations and options. Our intention with this paper is to present initial findings in exploring CASS's structure. Based on our early experimental analysis, presented in Section C.5, CASS seems to be a promising research direction for code similarity.

**An Important Weakness.** While CAST provides added flexibility over SPT, such flexibility may be misused. With CAST, system developers are free to add or remove as much syntactic differentiation detail they choose for a given language or given code body. Such overspecification (or underspecification), may result in syntactic overload (or underload) which may cause reduced code similarity accuracy over the original SPT design, as we illustrate in Section C.5.

## B    MODELS

In this section, we describe the models evaluated in our experiments other than MISIM-GNN, and discuss the details of the experimental procedure.

### B.1    MODEL: MISIM-BOF

The MISIM-BoF model takes a bag of manual features extracted from a CASS as its input. The features include the ones extracted from CASS trees, using the same procedure described in Aroma (Luan et al., 2019), as well as the entries in CASS GATs. As shown in Figure 5, the output code vector is computed by taking the elementwise mean of the feature embeddings and projecting it into the code vector space with a fully connected layer.

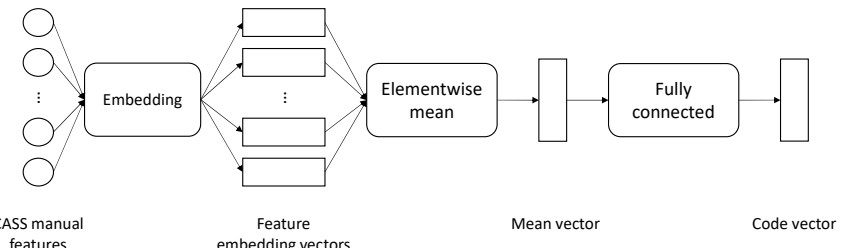

Figure 5: MISIM-BoF Architecture.

### B.2    MODEL: MISIM-RNN

The input to the MISIM-RNN model is a serialized representation of a CASS. Each CASS tree, representing a function in the program, is converted to a sequence using the technique proposed in (Hu et al., 2018). The GAT entry associated with a CASS tree is both prepended and appended to the tree's sequence, forming the sequence of the corresponding function. As illustrated in Figure 6, each function's sequence first has its tokens embedded, and then gets summarized to a function-level vector by a bidirectional GRU layer (Cho et al., 2014). The code vector for the entire program is subsequently computed by taking the mean and max pooling of the function-level vectors, concatenating these two vectors, and passing the resulting vector through a fully connected layer.

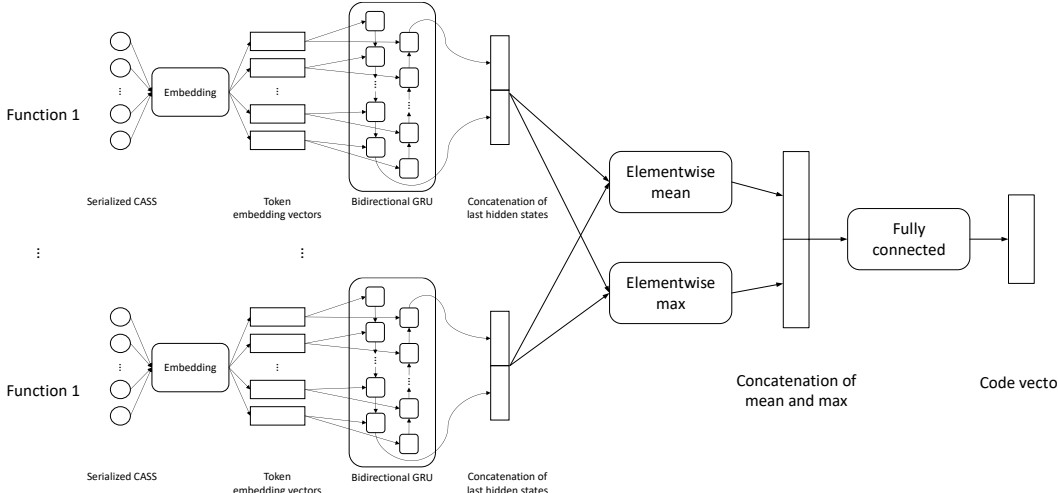

Figure 6: MISIM-RNN Architecture.

## C  EXPERIMENTAL DETAILS

### C.1  TRAINING PROCEDURE AND HYPERPARAMETERS

We use the AdamW optimizer with a learning rate of $10^{-3}$ (Loshchilov & Hutter, 2019). The training runs for 100 epochs, each containing 1,000 iterations, and the model that gives the best validation accuracy is used for testing.[8] The hyperparameters used for the Circle loss are $\gamma = 80$ and $m = 0.4$. For all of our MISIM models, we use 128-dimensional embedding vectors, hidden states, and code vectors. We also apply dropout with a probability of 0.5 to the embedding vectors. To handle rare or unknown tokens, a token that appears less than 5 times in the training set is replaced with a special UNKNOWN token.

### C.2  MODIFICATIONS TO CODE2VEC, CODE2SEQ, NCC, AND AROMA

To compare with code2vec, code2seq, NCC, and Aroma, we adapt them to our experimental setting in the following ways. The original code2vec takes a function as an input, extracts its AST paths to form the input to its neural network, and trains the network using the function name prediction task. In our experiments, we feed the AST paths from all function(s) in a program into the neural network and train it using the metric learning task described in Section 2.2.2. We make similar adaptions to code2seq by combining AST paths from the whole program as one input sample. Additionally, we replace the sequence decoder of code2vec with an attention-based path aggregator used in code2vec. NCC contains a pre-training phase, named inst2vec, on a large code corpus for generating instruction embeddings, and a subsequent phase that trains an RNN for a downstream task using the pre-trained embeddings. We train the downstream RNN model on our metric learning task in two ways. The first uses the pre-trained embeddings (labeled as NCC in our results). The second trains the embeddings from scratch on our task in an end-to-end fashion (labeled as NCC-w/o-inst2vec). For both code2vec and NCC, we use the same model architectures and embedding/hidden sizes suggested in their papers and open sourced implementations. The dimension of their output vectors (i.e., code vectors) is set to the same as our MISIM models. Aroma extracts manual features from the code and computes the similarity score of two programs by taking the dot product of their binary feature vectors. We experiment with both its original scoring mechanism (labeled: Aroma-Dot) and a variant that uses the cosine similarity (labeled: Aroma-Cos).

---

[8]We have observed that the validation accuracy stops to increase before the 100th epoch in all experiments.

## C.3 Evaluation Metrics

MAP@R measures how accurately a model can retrieve similar (or relevant) items from a database given a query. MAP@R rewards a ranking system (e.g., a search engine, a code recommendation engine, etc.) for correctly ranking relevant items with an order where more relevant items are ranked higher than less relevant items. It is defined as the mean of average precision scores, each of which is evaluated for retrieving R most similar samples given a query. In our case, the set of queries is the set of all test programs. For a program, R is the number of other programs in the same class (i.e., a POJ-104 problem). MAP@R is applied to both validation and testing. We use AP and AUPRG to measure the performance in a binary classification setting, in which the models are viewed as binary classifiers that determine whether a pair of programs are similar by comparing their similarity score with a threshold. AP and AUPRG are only used for testing. They are computed from the similarity scores of all program pairs in the test set, as well as their pair-wise labels. For the systems that require training (i.e., systems with ML learned similarity scoring), we train and evaluate them three times with different random seeds.

## C.4 MISIM Accuracy Results (in tabular form)

Table 5 shows the results of MISIM in comparison to other systems. Same results are presented in the graphical form in Figure 4.

Table 5: Code similarity system accuracy. Results are shown as the average and min/max values, relative to the average, over 3 runs. We had to make a few modifications to adapt code2vec, code2seq, NCC and Aroma to our experimental settings. Please refer to Appendix C.2 for details.

| Method | GCJ | | | POJ-104 | | |
|---|---|---|---|---|---|---|
| | MAP@R (%) | AP (%) | AUPRG (%) | MAP@R (%) | AP (%) | AUPRG (%) |
| code2vec | 7.76 (-0.79/+0.88) | 17.95 (-1.24/+1.76) | 25.48 (-7.37/+10.37) | 1.90 (-0.43/+0.38) | 5.30 (-0.80/+0.60) | 34.97 (-20.83/+15.10) |
| code2seq | 11.67 (-1.98/+1.73) | 23.09 (-3.24/+2.49) | 46.29 (-15.10/+12.02) | 3.12 (-0.45/+0.67) | 6.43 (-0.37/+0.48) | 54.97 (-7.15/+7.65) |
| NCC | 17.26 (-1.11/+0.57) | 31.56 (-1.11/+1.46) | 68.76 (-1.25/+2.46) | 39.95 (-2.29/+1.64) | 50.42 (-2.98/+1.61) | 98.86 (-0.20/+0.10) |
| NCC-w/o-inst2vec | 34.88 (-5.72/+7.63) | 56.12 (-7.63/+9.96) | 90.10 (-3.83/+4.49) | 54.19 (-3.18/+3.52) | 62.75 (-5.49/+4.42) | 99.39 (-0.22/+0.17) |
| Aroma-Dot | 29.08 | 42.47 | 82.03 | 52.09 | 45.99 | 98.42 |
| Aroma-Cos | 29.67 | 36.21 | 75.09 | 55.12 | 55.40 | 99.07 |
| MISIM-GNN | **74.90 (-1.15/+0.64)** | **92.15 (-0.97/+0.7)** | **99.46 (-0.08/+0.05)** | **82.45 (-0.61/+0.40)** | 82.00 (-2.77/+1.65) | 99.86 (-0.04/+0.03) |
| MISIM-RNN | 66.38 (-2.93/+2.68) | 85.72 (-1.19/+0.65) | 98.78 (-0.18/+0.1) | 74.01 (-2.00/+3.81) | 81.64 (-1.52/+2.72) | 99.84 (-0.03/+0.04) |
| MISIM-BoF | 72.21 (-0.64/+0.52) | 90.54 (-0.81/+0.56) | 99.33 (-0.07/+0.05) | 74.38 (-1.04/+1.04) | **82.95 (-0.70/+0.50)** | **99.87 (-0.01/+0.01)** |

## C.5 Experimental Results of Various CASS configurations

In this section, we discuss our experimental setup and analyze the performance of CASS compared to Aroma's simplified parse tree (SPT). In Section C.5.1, we explain the dataset grouping and enumeration for our experiments. We also discuss the metrics used to quantitatively rank the different CASS configurations and those chosen for evaluation of code similarity. Section C.5.2 demonstrates that, a code similarity system built using CASS *(i)* has a greater frequency of improved accuracy for the total number of problems and *(ii)* is, on average, more accurate than SPT. For completeness, we also include cases where CASS configurations perform poorly.

### C.5.1 Experimental Setup

In this section, we describe our experimental setup. At the highest level, we compare the performance of various configurations of CASS to Aroma's SPT. The list of possible CASS configurations are shown in Table 1.

**Dataset.** The experiments use the same POJ-104 dataset introduced in Section 4.

**Problem Group Selection.** Given that POJ-104 consists of 104 unique problems and nearly 50,000 programs, depending on how we analyze the data, we might face intractability problems in both computational and combinatorial complexity. With this in mind, our initial approach is to construct 1000 sets of five unique, pseudo-randomly selected problems for code similarity analysis. Using this approach, we evaluate every configuration of CASS and Aroma's original SPT on each pair of solutions for each problem set. We then aggregate the results across all the groups to estimate their

overall performance. While this approach is not exhaustive of possible combinations (in set size or set combinations), we aim for it to be a reasonable starting point. As our research with CASS matures, we plan to explore a broader variety of set sizes and a more exhaustive number of combinations.

**Code Similarity Performance Evaluation.** For each problem group, we exhaustively calculate code similarity scores for all unique solution pairs, including pairs constructed from the same program solution (i.e., program $A$ compared to program $A$). We use $G$ to refer to the set of groups and $g$ to indicate a particular group in $G$. We denote $|G|$ as the number of groups in $G$ (i.e. cardinality) and |g| as the number of solutions in group $g$. For $g = G_i$, where $i = \{1, 2, \ldots, 1000\}$, the total unique program pairs (denoted by $g_P$) in $G_i$ is $|g_P| = \frac{1}{2}|g|(|g| + 1)$.

To compute the similarity score of a solution pair, we use Aroma's approach. This includes calculating the dot product of two feature vectors (i.e., a program pair), each of which is generated from a CASS or SPT structure. The larger the magnitude of the dot product, the greater the similarity.

We evaluate the quality of the recommendation based on *average precision. Precision* is the ratio of true positives to the sum of true positives and false positives. Here, true positives denote solution pairs correctly classified as similar and false positives refer to solution pairs incorrectly classified as similar. *Recall* is the ratio of true positives to the sum of true positives and false negatives, where false negatives are solution pairs incorrectly classified as different. As we monotonically increase the threshold from the minimum value to the maximum value, precision generally increases while recall generally decreases. The *average precision* (AP) summarizes the performance of a binary classifier under different thresholds for categorizing whether the solutions are from the same equivalence class (i.e., the same POJ-104 problem) (Liu, 2009). AP is calculated using the following formula over all thresholds.

1. All unique values from the $M$ similarity scores, corresponding to the solution pairs, are gathered and sorted in descending order. Let $N$ be the number of unique scores and $s_1, s_2, \ldots, s_N$ be the sorted list of such scores.

2. For $i$ in $\{1, 2, \ldots, N\}$, the precision $p_i$ and recall $r_i$ for the classifier with the threshold being $s_i$ is computed.

3. Let $r_0 = 0$. The average precision is computed as:

$$AP = \sum_{i=1}^{N} (r_i - r_{i-1})p_i$$

### C.5.2 RESULTS

Figure 7a depicts the number of problem groups where a particular CASS variant performed better (blue) or worse (orange) than SPT. For example, the CASS configuration 2-0-0-0-1 outperformed SPT in 859 of 1000 problem groups, and underperformed in 141 problem groups. This equates to a 71.8% accuracy improvement of CASS over SPT. Figure 7a shows the two best (2-0-0-0-1 and 0-0-0-0-1), the median (2-2-3-0-0), and the two worst (1-0-1-0-0 and 1-2-1-0-0) configurations with respect to SPT. Although we have seen certain configurations that perform better than SPT, there are also configurations that perform worse. We observed that the configurations with better performance have function I/O cardinality option as 1. We also observed that the configurations with worse performance have function I/O cardinality option as 0. These observations indicates that function I/O cardinality seems to improve code similarity accuracy, at least, for the data we are considering. We speculate that these configuration results may vary based on programming language, problem domain, and other constraints.

Figure 7b shows the group containing the problems for which CASS achieved the best performance relative to SPT, among all 1000 problem groups. In other words, Figure 7b shows the performance of SPT and CASS for the single problem group with the greatest difference between a CASS configuration and SPT. In this single group, CASS achieves the maximum improvement of more than 30% over SPT for this problem group on two of its configurations. We note that, since we tested 216 CASS configurations across 1000 different problem groups, there is a reasonable chance of observing such a large difference *even if CASS performed identically to SPT in expectation.* We do not intend for this result to demonstrate statistical significance, but simply to illustrate the outcome of our experiments.

Figure 7c compares the mean of AP over all 1000 problem groups. In it, the blue bars, moving left to right, depict the CASS configurations that are *(i)* the two best, *(ii)* the median, and *(iii)* the two worst in terms of average precision. Aroma's baseline SPT configuration is highlighted in orange. The best two CASS configurations show an average improvement of more than 1% over SPT, while the others degraded performance relative to the baseline SPT configuration.

These results illustrate that certain CASS configurations can outperform the SPT on average by a small margin, and can outperform the SPT on specific problem groups by a large margin. However, we also note that choosing a good CASS configuration for a domain is essential. We leave automating this configuration selection to future work.

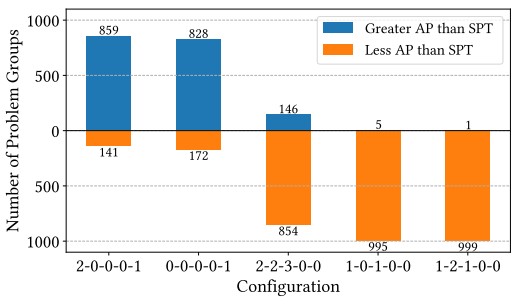

(a) Breakdown of the Number of Groups with AP Greater or Less than SPT.

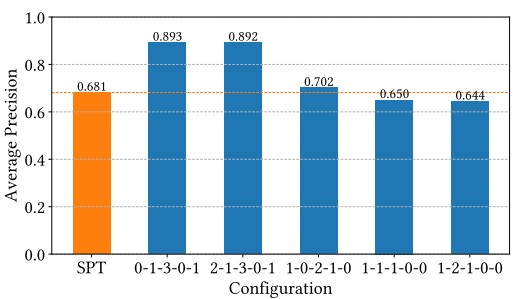

(b) Average Precision for the Group Containing the Best Case.

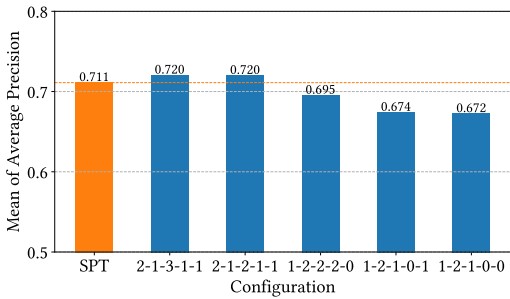

(c) Mean of Average Precision Over All Program Groups.

Figure 7: Comparison of CASS and SPT. The blue bars in (a) and (b), and all the bars in (c), from left to right, correspond to the best two, the median, and the worst two CASS configurations, ranked by the metric displayed in each subfigure.

### C.5.3 ANALYSIS OF CONFIGURATIONS

Figures 8a-8e serve to illustrate the performance variation for individual configurations. Figure 8a shows the effect of varying the options for the *node prefix label* configuration. Applying the node prefix label for the parentheses operator (option 2) results in the best overall performance while annotating every internal node (option 1) results in a concrete syntax tree and the worst overall performance. This underscores the trade-offs in incorporating syntax-binding transformations in CASS. In Figure 8b we observe that removing all features relevant to *compound statements* (option 1) leads to the best overall performance when compared with other options. This indicates that adding separate features for compound statements obscures the code's intended semantics when the constituent statements are also individually featurized.

Figure 8c shows that removing all features relevant to *global variables* (option 1) degrades performance. We also observe that eliminating the global variable identifiers and assigning a label to signal their presence (option 2) performs best overall, possibly because global variables appearing in similar contexts may not use the same variable identifiers. Further, option 2 performs better than the case where global variables are indistinguishable from local variables (option 3). Figure 8d indicates that removing features relevant to identifiers of *global functions*, but flagging their presence with a

special label as done in option 2, generally gives the best performance. This result is consistent with the intuitions for eliminating features of function identifiers in CASS as discussed in Section A.1. Figure 8e shows that capturing the input and output cardinality improves the average performance. This aligns with our assumption that function I/O cardinality may abstract the semantics of certain group of functions.

**A Subtle Observation.** A more nuanced and subtle observation is that our results seem to indicate that for each CASS configuration the optimal granularity of abstraction detail is different. For *compound statements* the best option seems to corresponds to the coarsest level of abstraction detail, while for *node prefix label*, *global variables*, and *global functions* the best option seems to corresponds to one of the intermediate levels of abstraction detail. Additionally, for *function I/O cardinality*, the best option has finer level of detail. For our future work, we aim to perform a deeper analysis on this and hopefully learn such configurations, to reduce (or eliminate) the overhead necessary of trying to manually discover such configurations.

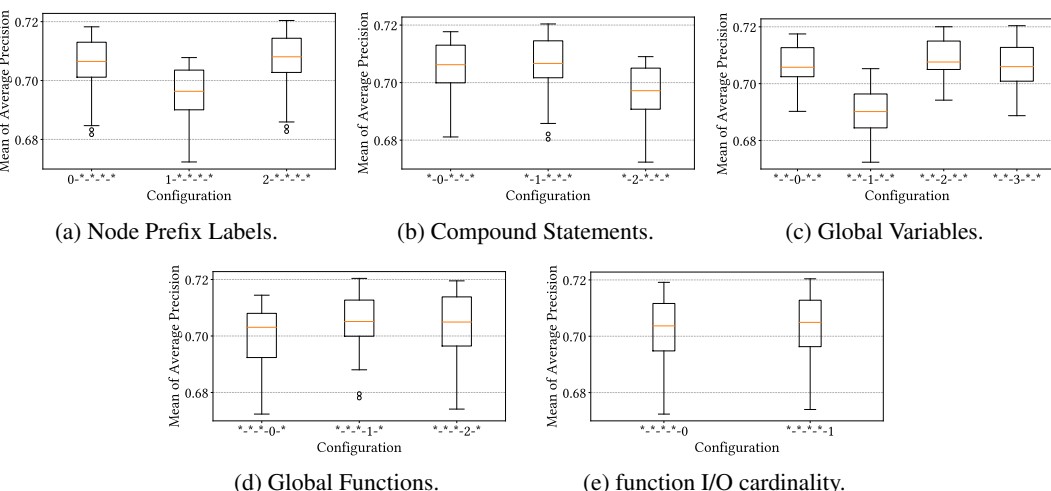

(a) Node Prefix Labels.

(b) Compound Statements.

(c) Global Variables.

(d) Global Functions.

(e) function I/O cardinality.

Figure 8: The Distributions of Performance for Configurations with a Fixed Option Type.

