# OpenReview forum: "(Updated submission 11/20/2020) MISIM: A Novel Code Similarity System"
_ICLR.cc/2021/Conference — Reject_

### Official Review · AnonReviewer1 · 2020-10-27
**lack of novelty and controlled comparison**

**Rating:** 4
**Confidence:** 5

**Review:**

This paper addresses the problem of code similarity detection and proposes an approach called MISIM.

Essentially, this paper proposes a tree representation, called context-aware semantic structure (CASS), which is very similar to the abstract syntax tree (AST). Then, different neural architectures (e.g., GNN or GRU) process the CASS and encode the program as a vector for similarity modeling.

Major concerns:

1. While the title of this paper claims "a novel code similarity system," I do not feel the model novel at all, as it just follows a standard Siamese structure.

The proposed CASS is very similar to abstract syntax tree, and there has been many structure variants proposed in previous work, like contextual flow graph, simplified parse tree. CASS is yet another, and I do not feel the need of CASS is well motivated.


2. Experimentally, the paper shows performance improvement on GCJ and POJ-104 datasets, compared with a few baseline systems, such as code2vec, NCC, and Aroma.

However, there's no controlled experiments on the choice of different structures, such as the AST and contextual flow graph. To be convinced, I expect

1) The authors show the results of controlled experiments with different structures.

2) The authors show the generality of the CASS in different tasks, such as clone detection.

3) The authors have a deeper analysis on CASS. What's the additional information encoded in CASS than a standard AST?

Minor:
    code2vec, NNC --> code2vec, NCC

---

> ### Author Response · Authors · 2020-11-20
> **Response to AnonReview1**
>
> Response to AnonReviewer1:
>
> Thank you for your review.
>
> >>> While the title of this paper claims "a novel code similarity system," I do not feel the model novel at all, as it just follows a standard Siamese structure. The proposed CASS is very similar to abstract syntax tree, and there has been many structure variants proposed in previous work, like contextual flow graph, simplified parse tree. CASS is yet another, and I do not feel the need of CASS is well motivated.
>
> Thank you for your comments. We agree with your observation, to an extent -- yes, there are certain similarities between AST and CASS, as you mentioned. However, we believe CASS has three distinct novelties in it that are possibly not representable in a classical AST representation. They are as follows.
>
> (1) When necessary, CASS can elide syntax entirely from the structure (making it a one-directional transformation, unlike an AST, which is bidirectional), rather than compounding into a compressed node. Doing this, we have found can help ensure that syntax doesn’t become a barrier (e.g., model overfitting) to identifying semantically similar code snippets that may be syntactically divergent. (2) CASS focuses on capturing structure in an attempt to lift out semantic code similarity, which may not always be possible from a purely syntactic representation. (3) Lastly, CASS can capture higher-order concepts that may not be easily identified from a purely syntactic viewpoint. One example of such a higher-order concept is around the **implications** of input/output function cardinality and how that impacts the semantics of a code snippet. This is currently captured in MISIM’s global attributes table (GAT), which, to our understanding, does not have an equivalent counterpart in an AST.
>
> In fact, as you had requested in your review, but may have not noticed in our appendix (we realize this is not required reading), we had done a controlled experiment on the GAT for the POJ-104 dataset before the ICLR submission. We cannot disclose the work here due to the anonymity policy. However, part of this experiment is included in the appendix of our submission, Figure 7 (a) shows that the best performing CASS configuration is 2-0-0-0-1 -- which uses the GAT. This seems to demonstrate to us that, at least for this dataset, GAT is improving accuracy for MISIM. In summary, it produces 859 groups that have better accuracy than the SPT. Our prior result on this experiment, without the GAT, shows that 2-0-0-0 is the best configuration and produces only 774 groups that have better accuracy than the SPT. This illustrates GAT provides 85 novel CASS configurations that improve semantic code similarity accuracy.
>
> >>> To be convinced, I expect: 1. The authors show the results of controlled experiments with different structures.
>
> Thank you for pointing this out; this is an excellent observation and we apologize for omitting this. We have since attempted to address this by performing a direct comparison of CASS to AST in a new Section 2.1.3 in the updated submission. In this section, we demonstrate CASS outperforms AST by 1.67X.
>
> >>> To be convinced, I expect: 2. The authors show the generality of the CASS in different tasks, such as clone detection.
>
> While we appreciate your perspective of considering different tasks for CASS/MISIM to be applied, we are not quite convinced that clone detection is actually a different domain than semantic code similarity. In short, we believe clone detection is a smaller subset of the space of semantic code similarity, because clone detection attempts to identify duplications of code within a corpus that are equivalent in either syntax or semantics (as we understand it). As such, this is a proper subset of semantic code similarity.
>
> >>> To be convinced, I expect: 3. The authors have a deeper analysis on CASS.
>
> We agree 100%. This is why we included a deeper analysis of CASS in the original submission’s appendix. We realize the appendix is optional reading, but we have done a fairly rigorous analysis of CASS in that section of the paper. It is our hope that you may reference it in a subsequent reading to then observe that this concern has already been addressed.
>
> If that analysis is still short of your request, please let us know, and we will be happy to provide as much of a deeper analysis as is possible before this rebuttal period ends. Thank you!

---

### Official Review · AnonReviewer4 · 2020-10-28

**Rating:** 5
**Confidence:** 5

**Review:**

### Summary ###

The paper presents a technique for code similarity. The paper combines two ideas: (1) a new “context-aware” representation for code which is used as a basis for computing code embeddings, (2) code similarity based on cosine similarity between the embeddings.

### Strengths ###

* The experimental results significantly improve previous work.

### Weaknesses ###

* Looks like an extension of AROMA, with a slightly more general approach for feature engineering, but no significant innovation beyond that.

* The code2vec baseline is old and weak (should have used code2seq).

### Questions for Authors ###

* You need to provide specific human customization per programming language. Can't this customization be learned itself? For example by using a Tree RNN to represent different abstractions of subtrees as needed?

* CASS seems to be closely related to an AST. In fact, the mapping of several syntactic elements (folding compound statements) could be viewed as a particular design choice when abstracting a syntax tree to an abstract syntax tree (in your case reflected in "node lables").

* The GAT includes a very shallow abstraction of function signatures. As such, I am curious about it's contribution to the final accuracy. Do you have an ablation study of the results obtained when GAT is omitted? I assume it has a rather different contribution across different languages?

* code2vec is a weak baseline. You should at least consider code2seq as it uses a significantly more powerful encoder.

* I wonder what would be the results using a textual sequence encoder as a baseline.


### Improving the Paper ###

* It would help to have a comparison of CASS+GAT to a simple AST-based approach. Keep your entire pipeline identical, including the linearization of the tree mentioned in B.2, and show what is the gain from the CASS per-language customization. I am sure that CASS+GAT would do better, but I conjecture that their contribution would be marginal in many mainstream languages.

* The comparison with Code2Vec is not clear to me. What do you mean by "we feed the AST paths from all function(s) in a program into the neural network and train it using the metric learning task described in Section 2.2.2.". Is this trained separately per program?

### Minor questions and comments ###

* This was a strange point to emphasize in Definition 1: "A child node is either an internal node or a leaf node. An internal node has at least one child node while a leaf node has no child nodes."

* Page 15, "pretended" -> prepended

---

> ### Author Response · Authors · 2020-11-20
> **Response to AnonReviewer4**
>
> Response to AnonReviewer4:
>
> Thank you for your review!
>
> >>> You need to provide specific human customization per programming language. Can't this customization be learned itself? For example by using a Tree RNN to represent different abstractions of subtrees as needed?
>
> We think this is an interesting observation and question. In fact, we have been building a secondary system, which we cannot go into too much detail here, that may allow such automatic abstraction representation learnings that could elide away the need for humans to provide the customizations that are currently necessary per programming language.
>
> If you would like, we can update the future work section of the paper, to make reference to this point. We wholeheartedly agree with this comment and think this is an excellent direction for future work.
>
> >>> CASS seems to be closely related to an AST. In fact, the mapping of several syntactic elements (folding compound statements) could be viewed as a particular design choice when abstracting a syntax tree to an abstract syntax tree (in your case reflected in "node lables").
>
> We agree with your observation, to an extent -- yes, there are certain similarities between AST and CASS, as you mentioned. However, we believe CASS has three distinct novelties in it that are possibly not representable in a classical AST representation. They are as follows.
>
> (1) When necessary, CASS can elide syntax entirely from the structure (making it a one-directional transformation, unlike an AST, which is bidirectional), rather than compounding into a compressed node. Doing this, we have found can help ensure that syntax doesn’t become a barrier (e.g., model overfitting) to identifying semantically similar code snippets that may be syntactically divergent. (2) CASS focuses on capturing structure in an attempt to lift out semantic code similarity, which may not always be possible from a purely syntactic representation. (3) Lastly, CASS can capture higher-order concepts that may not be easily identified from a purely syntactic viewpoint. One example of such a higher-order concept is around the **implications** of input/output function cardinality and how that impacts the semantics of a code snippet. This is currently captured in MISIM’s global attributes table (GAT), which, to our understanding, does not have an equivalent counterpart in an AST.
>
> To concretely demonstrate this point, we have added a new experimental result that demonstrates this empirically, comparing an AST to CASS. It can be found in section 2.1.3. It illustrates that CASS is 1.67X more accurate than AST for that experiment.
>
> >>> The GAT includes a very shallow abstraction of function signatures. As such, I am curious about it's contribution to the final accuracy. Do you have an ablation study of the results obtained when GAT is omitted? I assume it has a rather different contribution across different languages?
>
> In fact, yes, we have done an ablation study on GAT on the POJ-104 dataset before the ICLR submission. We cannot disclose the work here due to the anonymity policy. However, part of this experiment is included in the appendix of our submission, Figure 7 (a) shows that the best performing CASS configuration is 2-0-0-0-1 -- which uses the GAT. This seems to demonstrate to us that, at least for this dataset, GAT is improving accuracy for MISIM. In summary, it produces 859 groups that have better accuracy than the SPT. Our prior result on this experiment, without the GAT, shows that 2-0-0-0 is the best configuration and produces only 774 groups that have better accuracy than the SPT. This illustrates GAT provides 85 novel CASS configurations that improve semantic code similarity accuracy.
>
> >>> code2vec is a weak baseline. You should at least consider code2seq as it uses a significantly more powerful encoder.
>
> Thank you for this excellent suggestion. We have since added code2seq to our compared systems and now show that MISIM outperforms code2seq by upwards of 24.6x for POJ-104 and 6.4x for Google Code Jam. These results have been incorporated into the updated version of the paper (Section 4).
>
> >>> I wonder what would be the results using a textual sequence encoder as a baseline.
>
> Thank you for the suggestion. Unfortunately, due to the other large experiments and implementations that we felt were necessary to incorporate due to a majority of feedback from the other reviewers, we did not have time to address this. However, if our paper is accepted, we may have time to analyze this and incorporate the results in the camera-ready version.

---

### Official Review · AnonReviewer2 · 2020-10-28
**A new structual feature for source code**

**Rating:** 7
**Confidence:** 3

**Review:**

The main contribution of this paper is the CASS structual feature for source code and its high performance. The advantages of CASS are that it doesn't require compliation and even not completement, also the fast building time of the tree and the global attributes table, which lead the MISIM system can work in a live scenario.

Three kinds of deep learning models (GNN, RNN and BoF) follow the CASS representation to demonstrate its performance on two code similarity experiments. The scoring part uses a Siamese network to evaluate the different kinds of distance which is reasonable for solving this task.

Questions:
1. How CASS preserve the sementic information? It's more like syntax to me in its current form.
2. Could the CASS representation be extended to other software engineering jobs, such as code summarization, bug detection/repair or auto completion.
3. The phase 1 (CASS featurization) can be replaced by other code representation methods, such as CuBERT[1]. The power of CASS should be demonstrated through necessary comparsions.
4. Last, will the authors release their code to public?

[1] Learning and Evaluating Contextual Embedding of Source Code, ICML 2020

---

> ### Author Response · Authors · 2020-11-20
> **Response to AnonReviewer2**
>
> Response to AnonReviewer2:
>
> Thank you for your review!
>
> >>> How CASS preserve the sementic information? It's more like syntax to me in its current form.
>
> Our view is that CASS fundamentally preserves semantic information by first eliding away syntax that may limit semantic code information and instead to focus on higher-order representations and concepts such code structure and code mechanisms in code corpi.
>
> One such example of this, is CASS’s option that allows a user to differentiate the use of “()”, which in some languages (e.g., C/C++) can refer to either a function call or a mathematical operator. In comparison with the simplified parse tree (SPT) by Aroma, SPT does not provide such distinction, therefore that semantic information is lost at the node level.
>
> >>> Could the CASS representation be extended to other software engineering jobs, such as code summarization, bug detection/repair or auto completion.
>
> Yes, absolutely! In fact, we believe that CASS has the potential to be used for many other software engineering systems, such as code summarization (i.e., code comprehension), code reasoning and replacement, automatic bug detection, root cause identification for defects, autocompletion for both source lines and algorithms, etc. In fact, a core objective in designing CASS, is to enable its applicability for such various problems.
>
> >>> Last, will the authors release their code to public?
>
> Yes, we will! We have all the code included in our submission (except the new experiments, which we will update as soon as possible). We plan to have all code released and available to the public prior to the presentation of the paper if it is accepted to ICLR.

---

### Official Review · AnonReviewer3 · 2020-10-31
**Review for "MISIM: A Novel Code Similarity System"**

**Rating:** 5
**Confidence:** 4

**Review:**

This paper proposed a code similarity system called MISIM, which uses a context-aware semantic structure to represent code and calculate the similarity scores using a neural-based scoring algorithm. The context-aware semantic structure (CASS) is a structural representation of code specifically designed to lift semantic meaning from code syntax and provide an extensible representation that can be augmented. The neural-based similarity scoring part is implemented with deep neural networks, where authors tried 1). bag of features; 2). RNN, and 3). GNN. They found GNN performs the best. The experiment results show that MISIM can outperform code representing models like NCC and code2vec to detect code similarity.


Pros:
1.	The context-aware semantic structure for code representation is  interesting. The configurable code structure representation gives the model the flexibility to handle multiple tasks.
2.	This paper is well organized and easy to follow. The demonstration figures are clear and understandable.

Cons:
1.	This paper does not give detailed guidance on what configuration should be used for various circumstances. As shown in the appendix, different configurations will have a considerable influence on the experimental results. It is difficult for other researchers to apply CASS to different scenarios until a detailed configuration explanation is provided.

2.	Another weakness is that some important baselines are missing. This paper compares 3 different baselines: code2vec, NCC, and aroma, in which code2vec and NCC are designed for general code representation. Aroma is a code search system that locates code snippets in code corpus and makes code snippets recommendation. None of them are specially proposed for pair-wised code similarity comparison, as in this paper's experimental settings. There are many pieces of research about measuring code similarities such as CCLearner, CDLH, and DeepSim. Without comparison with those approaches, it is hard to convince others of the superiority of MISIM.


3.	The author emphasizes code similarity is now a first-order problem that must be solved in the introduction part, but the lack of relevant statics and examples makes this argument weak. Besides, the claim that AST representation will mislead code similarity systems into learning too much syntax but semantics also lacks solid evidence to support. Please provide some concrete examples that modeling those "syntax" in AST actually HURTS the performance, i.e., state-of-the-art methods learning from ASTs (e.g., the methods listed below) performs worse than the proposed method.

References:
[DeepSim] Zhao, G., & Huang, J. (2018, October). Deepsim: deep learning code functional similarity. In Proceedings of the 2018 26th ACM Joint Meeting on European Software Engineering Conference and Symposium on the Foundations of Software Engineering (pp. 141-151).

[CCLearner] Li, L., Feng, H., Zhuang, W., Meng, N., & Ryder, B. (2017, September). Cclearner: A deep learning-based clone detection approach. In 2017 IEEE International Conference on Software Maintenance and Evolution (ICSME) (pp. 249-260). IEEE.

[CDLH] Wei, H., & Li, M. (2017, August). Supervised Deep Features for Software Functional Clone Detection by Exploiting Lexical and Syntactical Information in Source Code. In IJCAI (pp. 3034-3040).

---

> ### Author Response · Authors · 2020-11-20
> **Response to AnonReviewer3**
>
> Response to AnonReviewer3:
>
> Thank you for your review!
>
> >>> The author emphasizes code similarity is now a first-order problem that must be solved in the introduction part, but the lack of relevant statics and examples makes this argument weak.
>
> While we believe this was fairly well substantiated, we have taken your comment under consideration and have toned down the language in our updated version of the paper (please see updated version).
>
> >>> It is difficult for other researchers to apply CASS to different scenarios until a detailed configuration explanation is provided.
>
> We agree that this may be challenging and it is an open problem for CASS research to learn the best configuration. Not only that, we believe that our current approach lays CASS as the foundation to specify the proper configurations to various problems, yet it can be further extended. In that sense, we expect that CASS options will grow not only from us, but hopefully from the community. Another potential future work is to automatically learn the proper CASS configuration for a given dataset. However, we believe all of this work is outside the scope of the initial MISIM architecture paper. These extensions are likely complete research papers in and of themselves.
>
> >>> the claim that AST representation will mislead code similarity systems into learning too much syntax but semantics also lacks solid evidence to support. Please provide some concrete examples that modeling those "syntax" in AST actually HURTS the performance, i.e., state-of-the-art methods learning from ASTs (e.g., the methods listed below) performs worse than the proposed method.
>
> Thank you for pointing this out. We have attempted to address this by performing a direct comparison of CASS to AST in Section 2.1.3. In this section, we demonstrate CASS outperforms AST by 1.67X.
>
> >>> Another weakness is that some important baselines are missing. There are many pieces of research about measuring code similarities such as CCLearner, CDLH, and DeepSim. Without comparison with those approaches, it is hard to convince others of the superiority of MISIM.
>
> To address part of your comment (and other reviewer’s), we have added a fourth system, code2seq, to our experiments bringing the experimental systems evaluation to five full state-of-the-art systems.
>
> However, we believe it’s important to note that there are many directly and tangentially related works in this space. We believe it may be unreasonable (and rather unrealistic) to expect any group of researchers to comprehensively compare against all of those systems as well as building their own novel system. Our latest version of the paper compares five state-of-the-art systems: Aroma (2019), code2vec (2018), code2seq (2019), MISIM (ours, 2020), and NCC (2018). Based on all the related works we compare against, not only do we believe that we empirically evaluate against more data, but we also seem to compare against more unique systems, comprehensively. Moreover, in terms of relative timeliness, all of the works we compare against have been published at tier-1 venues in the last two years. We believe this makes the comparison of their dataset size and empirical evaluations relevant as a metric to compare against our dataset size and empirical evaluation.
>
> While we appreciate your feedback on the other three systems you mentioned (CCLearner, CDLH, and DeepSim), we feel it’s important to note at least two things. (1) Two of those systems seem to be focused principally on clone detection, which we believe is not precisely the same problem as semantic code similarity. Instead, we see clone detection as a proper subset of semantic code similarity, where (near) equivalence for both semantic and syntax is found for two or more code snippets. Semantic code similarity, in our opinion, includes this analysis as well as the cases where semantic similarity is found even in the absence of syntactic similarity. (2) Two of the three systems you recommend are actually more dated than any of the ones we used, which, as reviewers have argued, perhaps make them less relevant. Following the consistency of other reviewer feedback, that would indicate we should not include them. That said, we have included citations to each of these works in our updated submission.
>
> Thank you for making us aware of them!

---

### Author Response · Authors · 2020-11-20
**To all MISIM reviewers**

Dear reviewers -

First, thank you for your detailed feedback and guidance. We have spent the last week and a half working to address as many of your comments and recommendations as possible. We have updated our submission with the following major changes:

(1) We have added new experimental data that demonstrates CASS is more accurate than AST (upwards of 1.67x) in Section 2.1.3.

(2) We have added new experimental data comparing code2seq to the other systems. MISIM remains the top performer against all four state-of-the-art systems, and outperforms code2seq by up to 24.6x for POJ-104 and 6.4x for Google Code Jam (Section 4).

(3) We have made a number of other revisions throughout the paper to address other comments such as missing citations, claims that needed to be toned down, typos, further clarifications, etc.

We apologize that it has taken us so long to put this rebuttal together. However, the implementation of two complete new systems and gathering all the necessary experimental data was more time consuming than we had anticipated. :)

We would be happy to hear any additional feedback you may have and are happy make any further changes (including new experiments assuming time permits) you may have before the rebuttal period ends.

We thank you again for your patience and your excellent feedback. We will subsequently be sending individualized responses to each reviewer.

Thank you again,
The MISIM Team

---

### Decision · Program_Chairs · 2021-01-07
**Final Decision**

**Decision:**

Reject

**Comment:**

This paper studies the problem of computing a similarity measure between two pieces of code. The main contributions are a configurable alternative (CASS) to abstract syntax trees (ASTs) for representing code and a model for embedding these structures within a Siamese net-like architecture. While parts of the ICLR community that make use of ASTs would likely find interest in the options provided by CASS and the associated experiments, the contribution is mostly around feature engineering of AST-like structures for one specific application, which is quite niche. The machine learning modeling appears fairly standard. Thus in total, I don’t see enough here to recommend acceptance.